# Nurses’ Workplace Perceptions in Southern Germany—Job Satisfaction and Self-Intended Retention towards Nursing

**DOI:** 10.3390/healthcare12020172

**Published:** 2024-01-11

**Authors:** Domenic Sommer, Sebastian Wilhelm, Florian Wahl

**Affiliations:** Technology Campus Grafenau, Deggendorf Institute of Technology, 94481 Grafenau, Germany; sebastian.wilhelm@th-deg.de (S.W.); florian.wahl@th-deg.de (F.W.)

**Keywords:** nursing, healthcare jobs, intention to leave, job satisfaction, employee retention

## Abstract

Our cross-sectional study, conducted from October 2022 to January 2023, aims to assess post-COVID job satisfaction, crucial work dimensions, and self-reported factors influencing nursing retention. Using an online survey, we surveyed 2572 nurses in different working fields in Bavaria, Germany. We employed a quantitative analysis, including a multivariable regression, to assess key influence factors on nursing retention. In addition, we evaluated open-ended questions via a template analysis to use in a joint display. In the status quo, 43.2% of nurses were not committed to staying in the profession over the next 12 months. A total of 66.7% of our surveyed nurses were found to be dissatisfied with the (i) time for direct patient care. Sources of dissatisfaction above 50% include (ii) service organization, (iii) documentation, (iv) codetermination, and (v) payment. The qualitative data underline necessary improvements in these areas. Regarding retention factors, we identified that nurses with (i) older age, (ii) living alone, (iii) not working in elder care, (iv) satisfactory working hours, (v) satisfactory career choice, (vi) career opportunities, (vii) satisfactory payment, and (viii) adequate working and rest times are more likely to remain in the profession. Conversely, dissatisfaction in (ix) supporting people makes nurses more likely to leave their profession and show emotional constraints. We uncovered a dichotomy where nurses have strong empathy for their profession but yearn for improvements due to unmet expectations. Policy implications should include measures for younger nurses and those in elderly care. Nevertheless, there is a need for further research, because our research is limited by potential bias from convenience sampling, and digitalization will soon show up as a potential solution to improve, e.g., documentation and enhanced time for direct patient time.

## 1. Introduction

### 1.1. Emerging Challenges in Nursing

Nurse satisfaction and retention are pivotal in healthcare globally. Exacerbated by pandemics, staff shortages, and high attrition rates, these factors underscore the need for robust nursing workforces [1,2,3]. In Europe, the aging population, multimorbidity, and decreasing amount of informal caregivers have increased the burden and reliance on professionals [4,5,6,7] Workloads and stress, particularly in Western Europe, have been escalating since COVID-19 [8]. A total of 18.4% of European nurses consider leaving nursing several times a month, thus indicating frequent quitting thoughts and highlighting the dire need for improved working conditions [9,10]. In Germany, these issues are pronounced, with one in five citizens being over 65 years old [3,11,12]. The pandemic has further highlighted the importance of a satisfied and retained nursing staff to manage growing healthcare needs [12,13,14]. German nurses are increasingly resigning, with 40% of nurses considering leaving monthly [15]. In addition, by 2023, Germany has predicted to face a shortfall of up to 500,000 nurses [9,16]. The German nursing sector sees high turnover rates, with nurses typically remaining in general nursing for six and in elderly care for ten years. High turnover rates have economic [17] impacts and affect the quality and continuity of care, thereby highlighting the need for strategies to retain nurses [18,19,20,21,22,23,24].

### 1.2. Lack of Transferable Research

Section 2 in the literature review identifies a research gap in nurse satisfaction and retention globally [1,25,26,27,28,29,30]. Contradictory findings and confounders in existing studies are noted. The variation in healthcare and working cultures necessitates national studies, as there is limited transferability across different systems [25,31,32].

There is a lack of comprehensive national data regarding nurses’ job satisfaction and self-reported retention [3,12]. Small sample sizes limit national studies such as the NEXT-Study or are outdated, thereby failing to reflect modern nursing practices [24,31,32].

In conclusion, there is a need for more comprehensive, holistic research to develop evidence-based interventions addressing nurses’ satisfaction and retention.

### 1.3. Contributions of Our Study

Despite the challenges in nursing, such as rising demand and outdated national retention data, our study aims to fill the research gap in understanding nursing conditions from the nurses’ perspective in Germany. Our research contributes in the following ways:(i)We present a national multifaceted understanding of the current job satisfaction of nurses and the factors that influence their self-stated nursing retention.(ii)We identify with our Bavarian-wide study important values and improvements regarding nurses’ working conditions, thereby enabling the evidence-based development of measures to enhance current nursing conditions.

## 2. Related Work

*Job satisfaction*, which is crucial in sociology, economics, and psychology, is defined as an emotional, (inter-) personal, and subjective state resulting from job appraisal and multidimensional factors, encompassing a spectrum from dissatisfaction to satisfaction [22,33,34,35,36]. *Employee retention* is linked to job satisfaction and can prevent turnover [10,35,37].

### 2.1. International Nursing Retention

Nurses, the largest group in healthcare, require enhanced job satisfaction and retention [26]. Studies analyze job satisfaction and retention factors globally [1,20,27,28,29,30,38,39,40,41,42,43]. As shown in Table 1, retention factors include (i) *personal attributes* (health, education, marital status, age), (ii) *job characteristics* (supportive leadership, autonomy, stress, and professional growth), and (iii) *organizational context* (wages, relationships, working environment, and employer location). Notably, organizational context and job characteristics are more impactful than individual factors [28,44]. Girma et al. [38], Cho et al. [43], and Aloisio et al. [29] found positive correlations between (i) *personal factors* in the form of higher educational attainment, being over 30 years old, and marital status with increased job satisfaction and retention. Older, more experienced employees with stronger employer relationships demonstrate a lower propensity to quit [41] Additionally, good health and physical fitness contributes positively [29]. However, personal factors are less influential than job and organizational aspects [29]. Regarding (ii) *job characteristics*, supportive leadership and autonomy positively correlate with satisfaction [27,28,29,30,39]. Niskala et al. [26] and Penconek et al. [27] emphasize leadership’s role, noting that nurses with leaders who foster shared decision-making, e.g., individual working patterns, are more satisfied [45]. Specchia et al. [39], Boateng et al. [20] and Hsu et al. [42] recognize leadership style as a key influencer on nurse satisfaction, which is further enhanced by development and promotion opportunities. Additionally, high job stress can lead to dissatisfaction and intentions to leave primarily when the workload is too demanding [20,27,29,40]. In (iii), the *organizational context*, Hsu et al. [42] found out that wages contribute to the intention to remain. Nonmonetary rewards, including good relationships with colleagues and patients, are essential [1]. Furthermore, the physical working environment, including access to supportive tools like lifter aids, enhances nurses’ contentment [28]. The rural or urban location of the employer also impacts satisfaction, with external factors affecting satisfaction more in urban areas due to the ease of changing employers [28].

### 2.2. National Nursing Retention

Nationally, job satisfaction and retention present significant challenges, with nursing retention declining over the past decade [13,25,46]. A total of 18.4% nurses in Germany consider quitting several times a month [10]. In addition, the average nursing tenure is less than ten years, thus indicating that many nurses leave nursing before retirement [9].Al Zamel et al. [1] found that retention factors vary between health systems, thereby underscoring the importance of national analysis. Roth et al. [32] suggest categorizing the influence on retention into three categories: (i) *financial and rewarding*, (ii) *professional*, and (iii) *personal* factors, which are consistent with international findings. Regarding (i) *financial aspects*, nurses’ dissatisfaction often stems from perceived inadequacies in (non-) monetary benefits compared to nurses’ workload and stress [25,32]. Balance between occupational efforts and rewards is a key factor for retention [15,23,24,47,48]. In the (ii) *professional factors*, recognition and involvement in decision-making processes have notable impacts [15]. The NEXT (nurses early exit) Study, which conducted an extensive investigation in the early 2000s, revealed that a higher intention to stay in the profession correlates with perceived leadership quality [10,24]. Moreover, increased work demands and strains, such as the emotional burden of end-of-life care, are influential, thereby highlighting the importance of (soft) social aspects like team culture, company atmosphere, and sufficient time for direct patient care [25,32,49]. Nonsufficient patient time and the connected unfinished care work represent an emotional burden, thus affecting retention [50]. Regarding (iii) *personal* factors, individual personalities, primarily emotional or affective, influences retention [51].

### 2.3. Implication of the Related Work to Our Research

As nursing satisfaction and retention are improvable, evidence-based measures to improve nursing conditions are needed [26]. Identifying different retention factors guided our study design, including all significant retention factors in our survey, build on existing measures. Nevertheless, there is a research gap with variations in influencing factors and limitations [15,24,25,32]. Niskala et al. [26] and Zamel et al. [1] suggest that countries’ health systems influence retention. National research has several limitations and needs to be updated, thereby indicating that our research is vital.

## 3. Methodology

From *October 2022 to January 2023,* we surveyed nurses in Bavaria, Germany.

### 3.1. Objectives and Research Questions (RQs)

Our single cross-sectional study is focused on two objectives: (i) to foster a broad national understanding of the prevailing satisfaction levels of nurses and the determinants that impinge on their retention in the profession and (ii) to procure primary data that can potentially provoke measures for improvement, which can be contextualized to contemporary challenges. In line with our objectives, we answer the following questions:RQ 1How satisfied are currently employed nurses with various work dimensions?RQ 2Which self-reported factors are considered important for nurses?RQ 3Which influencing factors on the intention to stay in the nursing profession (employee retention) can be explained in a statistical model?

### 3.2. Inclusion and Exclusion Criteria

We focused on *formal nurses*, synonymous with professional nurses spanning all institutional care forms, from long-term elderly care and outpatient services to clinical nursing. We excluded informal and family caregivers, as they did not align with our study’s purview. Eligibility criteria dictated that participants should be *currently employed* as nurses in Bavaria, Germany. Those who had left the nursing profession and those searching for nursing employment were deemed ineligible, mainly due to recruitment impracticalities.

### 3.3. Data Collection Process and Sampling

Given the unavailability of a random dataset, we adopted a convenience sampling strategy, spanning over 16 weeks from October 2022, via a LimeSurvey online survey. We partnered with Bavarian Health Regions (www.gesundheitsregionenplus.bayern.de, accessed on 3 January 2024) for our outreach, thereby leveraging their expansive network that covers nearly all 71 Bavarian counties. As part of our dissemination, institutions recruiting nurses across Bavaria were sent an introductory letter and a promotional poster. Limitations of our sampling are discussed in Section 5.

To address ethics and data protection, we implemented data privacy management and provided a self-ethical assessment. We anonymized personal data. An opt-in, with detailed information about the study and data protection (informed consent), was required for participation, and a cookie was embedded to prevent repetitive entries.

### 3.4. Data Collection Form (Online Survey)

Our online survey (Appendix F) was based on established instruments like the Index for Work Satisfaction (IWS) by Stamps et al. [52], the Minnesota Satisfaction Questionnaire (MSQ) [53], the Work-life Balance Scale [54], and the Nurse’s Retention Index [12,15,43,55]. We modified these validated tools to align with our research questions (RQs), thereby integrating aspects like organizational support and autonomy from the IWS, as well as scale and demographics from the MSQ. Our survey also encompassed career development and social needs for a comprehensive understanding of nurses’ situations. A pilot test with eight nurses ensured our survey’s reliability, thus leading to adjustments in clarity, wording, and structure based on their feedback. The survey structure assessed four thematic topics. We assessed the (i) *Importance of Work Dimensions* (employer policies, management practices, and leadership dynamics), followed by (ii) *Satisfaction with these Work Dimensions*, and (iii) *Job Commitment and Nurses Retention*. Finally, we assessed (iv) *Demographic Data*.

Our survey included (bi)nominal and ordinal variables on a four-point Likert scale ranging from ‘very important’ or ‘fully satisfied’ to ‘irrelevant’ or ‘not at all satisfied’, which are detailed in Appendix B. In addition, our online survey included five open-ended questions alongside close-ended ones, aimed at collecting qualitative data on nurses’ job situations and their tendencies towards job turnover.

### 3.5. Data Analysis of the Close-Ended Questions

We conducted a quantitative and qualitative mixed-method evaluation for joint displays. First, we explain our analysis of close-ended questions. Our quantitative analysis was tethered to evaluating the close-ended questions, which is explained in the following:(i)*Preparation and Descriptive Statistics:* Initially, we screened and removed all incomplete data per Döring et al.’s [56] guidelines. In each analysis, we present the total amount of participants (N). In preparation for regression analysis, we recorded variables into dummies and standardized continuous ones. Data analysis employed SPSS (Ver. 27), thus focusing on relationships between work dimension importance and satisfaction (RQ 1 and RQ 2). We maintained consistency for comparisons between RQ 1 and RQ 2. The initial steps involved *descriptive* statistics to examine all variables.(ii)*Bivariate Statistics:* We used *bivariate statistics* to explore differences between variables and to test differences between the dependent variable (DV) and independent variables (IVs) regarding RQ 3). The Mann–Whitney U test was performed for ordinally scaled, non-normally distributed variables. For normal scaled variables, the chi-square test was performed. For metric, nonparametric variables, the Mann–Whitney U test could also be applied. The significance level was assumed to be α=0.05 and an α=0.01 for highly significant. The significance level explains that the β error regarding the false null hypothesis rejection can be reduced with 95% probability.(iii)*Specification of the Dependent Variable (DV):* Relating to our dependent variable (DV), our regression was performed with one DV, with the intention of staying in nursing for the next 12 months. The DV had the options ‘yes, I plan to stay in nursing’, ‘maybe, if the conditions change’, or ‘no, I don’t plan to stay in nursing’. According to Shetty et al. [35] and Döring et al. [56], the retention variable (DV) was recorded binary with ‘yes, I intend to stay in the profession’ and ‘no, I intend to leave’ expressions (no and maybe if the conditions change in one category). The ‘maybe’ statement has uncertainties and is conditional on change so that it can be summarized.(iv)*Specification of Independent Variables (IVs):* The selection of IVs was literature-based, as described in Section 2.1. Table A4 in Appendix A shows the complete set of IVs, which are structured according to (i) personal, (ii) job, and (iii) organizational characteristics. For example, our IVs include gender, age, education, work commute, working hours, job experience, and satisfaction with dimensions like career opportunities, payment, working hours, leadership, service organization, or team cohesion. For further exploration, in Table 9, we additionally analyzed health as a potential contributor.(v)*Multivariable Binary Logistics Regression:* To examine factors influencing the nurse’s retention (RQ 3), we performed a stepwise *multivariable binary logistic regression analysis* with results shown in Table 10. IVs significantly related to the DV were included in the regression analysis, as listed in Table 2. As not all proposed IVs are significant, the model consists of IVs about (i) *Career and Training Opportunities*, (ii) *Working and Rest Times*, (iii) *Working Hours*, (iv) *Living Conditions*, (v) *Career Choice*, (vi) *Payment and Salary*, (vii) altruism, meaning to *Support People*, (viii) *Age*, and (ix) the *Work Area*.(vi)*Multivariate Model Diagnostics:* During the model specification, we examined the data to identify influential observations and potential multicollinearity among the independent variables (IVs). Predictors exhibiting multicollinearity were preemptively excluded to safeguard the model’s integrity, thus mitigating bias [57]. The IVs exhibited weak to moderate correlations among themselves, and no multicollinearity was identified. Regarding *model diagnostics*, the model’s predictive accuracy was assessed using measures such as the area under the receiver operating characteristic curve (ROC) and Nagelkerke’s R^2^, which were checked to determine a quality criterion. The ROC curve is a graphical representation of the model’s diagnostic ability, thus balancing sensitivity and specificity. Nagelkerke’s R^2^, a modification of the Cox and Snell R^2^, estimates the variance explained by the model, thereby serving as a goodness-of-fit measure. A higher Nagelkerke’s R^2^ indicates better model fit and predictive accuracy [58]. Our Nagelkerke’s R^2^ was with 0.38% acceptable, which is discussed more extensively under Section 5, including specific constraints.(vii)*Result Interpretation:* We *interpreted* the results according to the survey’s framework and our RQs. The final report presents all significant estimated coefficients, odds ratios, standard errors, and *p*-values, thereby offering an overview of the findings.

### 3.6. Data Analysis of Open-Ended Questions

In tandem with evaluating close-ended questions, we deployed a *template analysis* of open-ended questions in our cross-sectional study. The qualitative evaluation was done to enrich the statistical analysis, to support answering RQ 1 and RQ 2 (Section 3.1), and to capture nurses’ perceptions regarding the job and organizational characteristics. The survey incorporated unrestricted free-text sections, thereby enabling participants to proffer detailed reflections. Participants were asked to share comments and insights, primarily satisfaction or improvement suggestions, about various dimensions. First, we assessed (i) *Employer and Organizational Policies*, following (ii) *Nursing and Organization of Care*, (iii) *Social Aspects* (incl. Patient Relations), and (iv) *Personal Conclusion* (incl. Retention).

We employed a modified template analysis method with *MaxQDA* (Ver. 20), thereby combining approaches from King et al. [59] and Brooks et. al. [60], as well as the qualitative content analysis techniques of Braun et al. [61] and Mayring et al. [62]. The following consecutive steps visualize our qualitative analysis approach:(i)*Familiarization and Preliminary Coding:* Initially, personal data, e.g., institution names, were removed to ensure data protection. We conducted multiple readings of all text for understanding, which was followed by initial coding based on first impressions.(ii)*Thematic Clustering:* Themes emerging from the data were organized into clusters with clear inclusion criteria, thereby forming thematic meaningful groups.(iii)*Template Design:* An initial coding template, informed by data impressions and the survey’s structure, was developed using MaxQDA for tagging and categorization. MaxQDA helped us by organizing tags, subcategories, and supercategories. We applied, tested, and modified our template, thus reorganizing and adding themes.(iv)*Finalization and Multiple Coding:* The coding was finalized and applied to all data, with passages categorized into relevant themes after two comprehensive reviews.(v)*Quality Assurance:* A second researcher (study author) reviewed the data and coding to ensure consistency and reliability in the coding process.

Our qualitative analysis, conducted on extensive responses from 950 nurses, with each providing at least one free-text comment, revealed key themes across five thematic categories: ‘Overall working conditions and policies’, ‘Leadership and Line Managers’, ‘Regulatory and Given Framework Conditions’, ‘Self-Esteem and Nursing Profession’, and ‘Retention to the Nursing Profession’. Each category was further divided into subcategories, with the most frequent ones detailed in Table 11. Most common subcategories primarily include reasons for leaving nursing and insights on staffing, nursing ratios, and duty scheduling. The complete classification system and qualitative data are available as individual materials upon request. In Section 4.6, we report the results using German translated quotes. We also compare qualitative conclusions with quantitative assumptions.

## 4. Results

Next, we present the demographics, nurses’ satisfaction, and job retention analysis.

### 4.1. Demographics and Characteristics of the Study Population

Our N = 2572 nurses, which were mainly from Bavaria (Germany), who participated in our study. ZIP codes starting with 94 were most common as a residential region of the surveyed nurses, thus meaning Lower Bavaria was represented with 598 (23.3%) participants. In contrast, other ZIP codes had a share of less than 12.1%. A total of 81.4% (n = 1707) of the nurses surveyed were women, with 60.5% (n = 1355) having at least one child. Age decentiles were evenly distributed, except for younger nurses under 20 years at 3.6% (N = 80) and those over 60 at 6.4% (N = 143). Living situations mostly involved having another person in the household at 65.1% (N = 1433).

When looking at *employment settings*, 57.9% (N = 1170) of our study participants worked in inpatient hospital care, followed by long-term care at 23.9% (N = 484), outpatient care at 12.1% (N = 305), and facilities for disabilities at 3.1% (N = 62). It was found that 80.7% (N = 1789) had a work commute of less than 30 min, thereby showing that nursing is a job near home. Regarding professional experience 54.4% (N = 1201) had accumulated 15 years or more, while the remaining 45.6% (N = 1008) had less than 14 years. However, the long-standing experience did not necessarily mean a long tenure with their current employer. A total of 34.8% (N = 771) had been with their current employer for under five years, with 31.2% (N = 689) having over 15 years of company affiliation. A total of 28.3% (N = 629) of the surveyed nurses held a leadership position. As per weekly working hours, 57% (N = 1270) indicated a working time of more than 35 h, thereby showing a predominance of full-time employment.

Alongside the 2572 respondents to closed-ended questions, 950 nurses answered open-ended questions, which are detailed under Section 4.6. A detailed demographic breakdown of our study population is found under Appendix A, thereby highlighting no marked disparities between respondents of close-ended and open-ended questions.

### 4.2. Job and Organizational Characteristics of Nursing

We assessed nurses’ job satisfaction and the importance of job components, regarding the survey’s order and Table 1. To ease understanding, we summarized the four-point Likert scale importance and satisfaction into bivariate scales. For example, ‘satisfied’ was aggregated from ‘fully’ and ‘rather satisfied’, and dissatisfaction was aggregated vice versa.

#### 4.2.1. Employer and Organizational Policies

In our analysis of employer and organizational policies, presented in Table 3, we adopted a unified approach in table formats to synthesize the perceived *importance* and *satisfaction* for clearer understanding and comprehension. The following enumeration summarizes the results in this dimension. This kind of enumeration is used in various subchapters, visualizing the results more effectively. For most nurses, the following employer and organizational policies were found to be *highly relevant*:(i)‘Payment and Salary’ with 98.3% (N = 2503),(ii)‘Leadership Recognizes Suggestions’ with 97.3% (N = 2449),(iii)‘Work-Family Reconciliation’ with 97% (N = 2491),(iv)‘Work Promotes Health’ with 94.8% (N = 2438).

Regarding the self-reported *satisfaction,* the *most dissatisfying* dimensions were:(i)’Co-Determination Rights’ with 52.3% (N = 1198),(ii)’Payment and Salary’ with 52.1% (N = 1196),(iii)‘Work-Family Reconciliation’ with 49.6% (N = 1139).

Analyzing employer and organizational policies revealed disparities between what nurses deem relevant and their corresponding satisfaction levels, thereby underscoring areas for potential improvement. While ‘Payment and Salary’ were important for over 90%, over half of the nurses were dissatisfied with this aspect. Similar trends were observed in ‘Co-determination rights’ and ‘Work-Family reconciliation’, indicating that these are priority areas where targeted measures of healthcare organizations could foster nurse retention. The high value placed on these factors, coupled with substantial dissatisfaction, suggests that improvements in these areas could enhance nurse retention. In this section, we focus on the objective result presentation. The impact of the importance of work aspects and the actual satisfaction, as well as measures, will be discussed more deeply in Section 5.

#### 4.2.2. Nursing and Care Organization

Nurses’ answers regarding their nursing organization are shown in Table 4.

For most nurses the following nursing organization characteristics were *highly relevant*:(i)‘Time for Patient Care’ with 99.5% (N = 2549),(ii)‘Plannable Working & Rest Times’ with 97.3% (N = 2491),(iii)‘Working and Auxiliary Tools’ with 95.9% (N = 2458),(iv)‘Reliable Service Organization’ with 95.2% (N = 2446).

*Dissatisfaction* prevailed (>50%) regarding the following dimensions in this section:(i)‘Time for Patient Care’ with 66.7% (N = 1533),(ii)‘Reliable Service Organization (e.g., with few stand-ins)’ with 58.3% (N = 1340),(iii)‘Nursing Documentation’ with 54.9% (N = 1252).

The highest disparity between importance and satisfaction was found in ‘Time for Patient Care’, ‘Reliable Service Organization’, and ‘Plannable Working & Rest Times’.

#### 4.2.3. Social Aspects in the Scope of Nursing

Regarding nurses’ social aspects, we list the self-reported perception in Table 5.

For most nurses, the following social aspects in nursing were *highly relevant*:(i)‘Team Cohesion (e.g., Relationship to Colleagues)’ with 97.7% (N = 2508),(ii)‘Supporting People in Tough Situations’ is considered crucial by 96.0% (N = 2458),(iii)‘Relationship with Managers’ is deemed highly significant by 89.8% (N = 2300).

Regarding the social aspects, satisfaction prevailed. *Dissatisfaction* was as follows:(i)‘Relationship with Managers’ with 40.0% (N = 918).(ii)‘Supporting People in Tough Situations’ with 37.8% (N = 847),(iii)‘Team Cohesion (e.g., Relationship to Colleagues)’ with 21.3% (N = 490).

As the results show, three-quarters were satisfied with the team, and satisfaction prevailed in social aspects. Nevertheless, our results indicate a disparity between the relevance and satisfaction of social aspects. Social aspects were deemed to be highly important for nurses. The highest discrepancy can be seen in the ‘Relationship with Managers’, which, despite being regarded as important by 89.8% (N = 2300) of the respondents, satisfied 60% (N = 1377) of respondents. Similarly, ‘Supporting People’ and ‘Team Cohesion (e.g., Relationship with Colleagues’ showed a disparity between importance and satisfaction.

#### 4.2.4. Summary of Job and Organizational Characteristics

The revealed disparities offer a proposal for a guideline to prioritize measures and are summarized in Table 6. Our results highlight that monetary factors are not the most important and common points of dissatisfaction. The top five values respondents considered most important were ‘Time for Patient Care’ (99.5%), ‘Payment and Salary’ (98.3%), ‘Team cohesion (97.7%), ‘Leaders Recognize Suggestions’ (97.3%), and ‘Plannable Working & Rest Time’ (97.3%). Satisfaction levels ranged from 33.3% for ‘Time for Patient Care’ to 59.1% for ‘Leaders Recognize Suggestions’. The top five dimensions that the respondents considered most dissatisfying were ‘Time for Patient Care’ (66.7%), ‘Reliable Service Organization’ (58.3%), ‘Nursing Documentation’ (54.9%), ‘Co-Determination Right(s)’ (52.3%), and ‘Payment and Salary’ (52.1%). A notable gap exists between nurses’ job satisfaction and their perceived importance of dimensions, particularly in ‘Time for Patient Care’.

### 4.3. Description of the Individual’s Conclusion, including the Occupational Decision and Retention

Nurses were asked for their *happiness with their occupational decision* and if they would enter nursing again. A total of 27.6% (N = 632) said they would definitely choose to nurse again, while 55.3% (N = 1267) would reconsider only if work conditions improved, and 17.1% (N = 393) regretted becoming nurses (Table 7).

Regarding the *intention to stay in nursing*, which was later recoded to our dependent variable (DV), Table 8 reveals the results. A total of 56.8% (N = 1299) said they intend to remain in nursing for the next 12 months, while 32.7% (N = 747) said they are open to staying if conditions improve, thereby suggesting a potential retention challenge. Additionally, 10.5% of nurses (N = 240) expressed a definite intention that they are sure about staying in the profession in the next 12 months.

### 4.4. Health-Related Feasibility of Continued Nursing

Table 9 visualizes the self-reported health situation as a potential retention limitator.

Among the nurses surveyed, 14.7% (N = 300) confidently expressed their health-related ability to continue nursing, thus attributing this to good health. In contrast, 53.3% (N = 1086) indicated that their decision to stay is conditional and dependent on improvements in their occupational situation, thereby highlighting the uncertainty and potential health risks in the current nursing setting. Additionally, 31.9% (N = 650) felt that continuing in nursing was unfeasible due to health concerns. While some nurses said they are committed to staying, a significant portion needed clarification, with their decisions affecting their working conditions. The findings reveal that health is a pivotal factor in nursing retention, with about one-third viewing it as a barrier to working in the profession in the next year. The impact of health on the feasibility of continuing in nursing and potential solutions are discussed in greater depth in Section 5.

### 4.5. Logit Model: Influence Factors on the Intention of Staying in Nursing Profession

For answering RQ 3, we conducted a stepwise multivariable binary logistic regression analysis. Regarding the *integration of variables* we followed a stepwise approach, with the model building detailed in Section 3. The dependent variable in our regression analysis was binary-encoded, thus targeting the response category ‘Yes, I plan to stay’. IVs were integrated due to significance in the existing literature and substantiated by preliminary bivariate statistics. We assessed multicollinearity among IVs, with weak to neglectable correlations being found. Finally, nine significant IVs were incorporated into our model, including (i) ‘Age’, (ii) ‘Living Conditions’, (iii) ‘Work Area’, (iv) ‘Working Hours’, (v) ‘Career Choice’, (vi) ‘Career and Training Opportunities’, (vii) ‘Payment and Salary’, (viii) ‘Working and Rest Times’, and (ix) ‘Supporting People’.

Regarding the *model quality*, our model in Table 10, was found to be statistically significant (X2 (10) = 568.09, *p* < 0.001, N = 1711). Our model accounted for approximately 38.0% of the variance in nurse retention (as per Nagelkerkes R^2^ determination coefficient), and it successfully classified 73.3% of instances correctly, thus attesting to robustness.

*Model Explanation:* Our model in Table 10, explains the impact of factors ranging from personal aspects to job satisfaction elements on nurses’ decisions to stay in nursing. Our model delivers the basis for evidence-based measures for healthcare policy, thereby effectively bridging the research gap stated in Section 2 and improving practical nurses’ conditions. We recommend considering the model factors as policy implications. Further discussion on the limitations and implications are presented in Section 5.

(i)*Age:* Regarding demographics, age positively correlated with retention. With every unit increase in age, the odds of retention were enhanced by 0.21 points (OR = 1.23, 95% CI = 1.13–1.35), thereby suggesting that older nurses are more likely to remain.(ii)*Living Conditions:* Nurses living separately or alone demonstrated a higher propensity for job retention than other living condition categories (OR = 2.59, 95% CI = 1.37–4.89), thereby implying that certain personal circumstances might bolster job retention.(iii)*Work Area:* Conversely, employment in stationary elder care was negatively associated with retention (OR = 0.71, 95% CI = 0.53–0.95), thereby suggesting higher attrition and hinting at potential systemic issues in elderly care that require attention.(iv)*Working Hours:* Working hours were shown to influence the intention to stay (OR = 1.20, 95% CI 1.01–1.42), thereby meaning higher working hours facilitate retention.(v)*Career Decision:* When considering the happiness of the occupation decision, the reentry into nursing, those expressing willingness to do so had significantly higher odds of retention (OR = 23.32, 95% CI = 15.02–36.20). Moreover, nurses indicating potential reentry contingent upon changes in conditions showed a greater likelihood of job retention than those unwilling to re-enter (OR = 2.98, 95% CI = 2.18–4.07). These findings underline the impact of individuals’ attitudes toward nursing retention.(vi)*Career Development Opportunities:* Satisfaction with ‘Career and Training Opportunities’ (OR = 1.33, 95% CI = 1.15–1.54) was related to higher retention.(vii)*Satisfaction with Payment and Salary:* Satisfaction-related factors showed a strong positive association with nursing retention. Satisfaction with a salary positively influenced retention decisions (OR = 1.23, 95% CI = 1.00–1.27). This item highlights the relevance of financial satisfaction and, more generally, the financial security situation.(viii)*Satisfaction with Working and Rest Times:* Satisfaction in working and rest times contributes to increased retention (OR = 1.28, 95% CI = 1.08–1.51). This factor emphasizes the need for a work-life balance in promoting retention.(ix)*Supporting People in Tough Situations:* Contrarily, supporting people in challenging situations was negatively associated with retention (OR = 0.72, 95% CI = 0.61–0.85).

### 4.6. Evaluation of Open-Ended Questions: Nurses’ Perspective(s)

In addition to our quantitative evaluation, we performed a template analysis. Not all nurses responded to open-ended questions. N = 950 nurses wrote one or more comments to the open-ended questions. Most nurses’ statements concerned, as visualized in Table 11, the area of ‘Overall Working Conditions & Policies’, thereby constituting 44.87% (N = 988) of responses. The second most prominent theme was ‘Regulatory & Given Framework Conditions’, which comprised 17.48% (N = 385) of responses. ‘Retention to the Nursing Profession’ accounted for 15.26% (N = 336), ‘Self-Esteem and Nursing Profession’ for 12.44% (N = 274), and ‘Leadership & Line Managers’ for 9.95% (N = 219) of all responses.

#### 4.6.1. Overall Working Conditions and Policies

We integrated illustrative quotes and structured them according to Table 11. Table 12 summarizes qualitative findings leading to practical and policy-making implications.

In detail, the following qualitative results could be analyzed:(i)*Staffing and Nursing Ratios:* Existing working policies lead nurses to perceive understaffing and task overabundance. Nurses are usually busy and face emotional strain, as the following quote shows: *“The dire nature of our situation is that we can’t spend even five min with a dying person begging for companionship” (Text 4b, para. 139).* Furthermore, this is described as a dilemma and needs to change with the establishment of minimum staffing: *“The solution to our dilemma would involve a new calculation of the nursing staff ratio; distributing the workload across more shoulders” (Text 2b, para. 267).*(ii)*Duty Scheduling and Working Hours:* Participants state that work schedules are not satisfactory, especially because of long consecutive services: *“Many of us work up to 12 consecutive days and are still required to cover additional shifts. This is physically and mentally demanding” (Text 1a, para. 346).* Duty Schedules need restructuring to be more reliable and should have a limit of consecutive shifts (Text 1a, para. 472). *“It is necessary to cut back on consecutive work days. No more than ten days in a row should be the norm” (Text 2b, para. 91).* Unreliable and often changing plans are demanding: *“Many taking over unplanned shifts and being phoned when you have time off is ruining things. In addition, rotating shifts break you down” (Text 1a, para. 472).* In addition to the relevance of downtimes, nurses desire more family-friendly working plans, with extraordinary mother shifts and reduced instances of spontaneous shift coverage (Text 5c, para 10). Family-friendliness and work-life balance are important concerns, as the following nurse states: *“Reconciling work hours with family life is challenging due to shift work. […] Work on weekends and holidays also impacts our social life significantly” (Text 5c, para. 305).*(iii)*Payment and Compensation:* Participants voiced a need for improved compensation for holiday work, on-call services, sick leave cover and additional shifts, thereby alluding to what they described as a *“gratification crisis” (Text 2b, para. 524).* The following quote visualizes the importance of rewards for long shifts: *“An appropriate payment should be in place for on-call duties that require one to stay in the hospital for 24 hours. Currently, one might work up to 24 hours but only receive 60% of the wage. This is an outdated practice that would be unthinkable in other sectors” (Text 1a, para. 209).* As the *“payment doesn’t reflect the mental and physical effort involved in nursing” (Text 5c, para. 68).*, nurses called for measures regarding better remuneration that reflects nurses’ work demands.(iv)*Health-Related Challenges:* Nurses emphasized the physical and mental stressors: *“Nursing is fulfilling, but burnout is pre-programmed under the present conditions. Work on a piecework basis, no time for patients, more and more patients per nurse, alone at night with 34 patients, often only two during the day. The motto is to get through the shift without anyone dying” (Text 5c, para. 402).* Nurses express an emotional toll and frustration at being unable to fulfill their roles (Text 5c, para. 291). Nurses yearn to provide *“human attention” (Text 5c, para. 404)* to patients. Patient time is nurses’ vital concern.The growing mental strain is compounded by the lack of support strategies and increasing patient aggression (Text 4b, para. 70), thereby leading to physical and psychological impacts on nurses. *“The rising aggression among patients is alarmingly high, so it’s not uncommon to go home with bruises or, even worse, being unable to switch off after work because you just can’t decompress assaults” (Text 5c, para. 291)*. *“Apart from constant verbal or physical assaults, the psychological strain caused by screaming and constant ringing, neglected patients who refuse personal hygiene and medication, is an immense burden, as is the stress caused by under-staffing and time pressure” (Text 4b, para. 154).*Physically, nursing often results in musculoskeletal disorders, with nurses feeling unsupported (Text 7c, para. 276). Better occupational health management is needed, including health promotion, reintegration programs, and age-appropriate workplace designs, which are currently lacking (Text 1a, para. 470, 54). The *“intense circumstances of the pandemic” (Text 4b, para. 144)* intensified the challenges (Text 5c, para. 81). *“Many nurses feel exhausted” (Text 4b, para. 144).* Additionally, a strong sense of duty sometimes results in nurses working while ill (Text 5c, para. 404). There is a call for more *“back-friendly work, grief recovery, and (…) support” (Text 7c, para. 370).*(v)*Debureaucratization:* Nurses are concerned about the rising bureaucracy in their profession: *“Year by year, we spend more time on largely meaningless documentation, with less and less time for nursing and care” (Text 1a, para. 401).* Documentation seems to be time-consuming and frustrating due to the detraction from direct patient care (Text 1a, para. 166). Solutions for bureaucratization include assistance with documentation, streamlining documentation processes, and using digital tools to save time (Text 1a, paras. 166–168). *“Nurses should be able to focus on their roles as nurses rather than secretaries or accountants” (Text 1a, paras. 166–168)*.(vi)*Corporate and Team Culture:* Concerning corporate culture, nurses noted that a lack of accountability in addressing issues and grievances was common (Text 4b, para. 208). They called for improved communication and informal meetings with leaders, departments and colleagues (Text 4b. para. 153). Furthermore, nurses acknowledged the benefits of effective team collaboration (Text 4b, para. 142).

#### 4.6.2. Regulatory and Given Framework Conditions and Liabilities

Many respondents see potential in digitalization, thereby potentially fulfilling the desire to spend more direct patient time. Table 13 illustrates our framework findings:

In detail, qualitative results regarding the framework conditions were analyzed:(i)*Digitalization:* Current digital healthcare solutions are limited by operation speed, updates, glitches, unreliable connections, and a lack of interfaces and software that only encompass part of nursing processes and waste time (Text 1a, para. 265). Despite seeing the digitalization potential for efficiency, nurses felt that more investments were needed. Nurses prefer one-parent systems: *“We need meaningful digitization with functional programs and interfaces, not a multitude of individual software” (Text 1a, para. 150)*. Furthermore, *“training and user-friendliness (Text 1a, para. 296)”* and that *“digitalization must become easier” (Text 3b, para. 117)* were essential to nurses.(ii)*Specifications and Inspections:* Nurses felt that political change was necessary, with many advocating for earlier retirement: *“How can I provide quality care if I’m expected to work until 67? We might end up needing assistance ourselves while trying to help” (Text 7c, para. 42)*. Nurses also want to change their representation through a chamber (Text 1a, para 288). Concerning the pandemic, strict rules, including mandatory vaccination and mask-wearing, were criticized (Text 1a, para. 187). Some nurses stated that wearing FFP2 masks nonstop was an *“equivalent to a physical assault” (Text 1a, para. 197)* and emphasized the demanding conditions during COVID-19. Backed by pandemic experiences, nurses predominantly have a negative view towards privatizing healthcare facilities, thus leading to statements like the following: *“The privatization of healthcare must be stopped. Health and care insurance money can’t lead to profits and shareholder disbursements up to 15%, besides on [nurses] back” (Text 1a, para. 180).* Nurses emphasize the importance of public welfare, thus seeing healthcare as a *“state duty” (Text 1a, para. 180).*(iii)*Training and Education:* Training and education are essential and must be reflected in nursing. Nurses want better support and supervision for trainees, expanded professional competencies, and increased focus on practical training (Text 5c, para 56). There should be more career opportunities, as shown in the following quote: *“Nursing should follow a U.S.-like professionalization model, with refined degrees leading up to physician assistant or physician” (Text 5c, para. 56).* Nurses also highlighted the importance of *“superior training” (Text 1a, para. 528)* and *“language […] courses and examinations” (Text 2b, para. 390)*. Apprentices should be mentored and guided, trainees *“should not manage a ward independently and always have a contact person on site” (Text 1a, para. 528).*Nurses also mentioned the relevance of lifelong learning, including the need for ongoing support to enhance their proficiency (Text 1a, para 31). Suggestions to advance nursing education ranged from extending the duration of training to developing academic programs paralleling those in the medical field (Text 1c, para 342).(iv)*Working Aids and Equipment:* Nurses raised worries about aids such as standing aids, slings, and boards for patient transfers, noting *“simple equipment like toilet chairs or wheelchairs are often outdated, broken, and insufficient” (Text 3b, para. 322). “More and better provision of nursing aids is necessary for the raising number of elderly” (Text 1a, para. 434).* In addition, nurses highlighted the facilities needing *“ larger patient rooms that promote freedom of movement [and] mobilization” (Text 3b, para. 71).* In addition to more spacious facilities, *“dedicated administrative areas” (Text 2b, para. 511)* and suggestions to install automatic sliding doors to improve accessibility were mentioned.

#### 4.6.3. Self-Esteem and Nursing Profession Perception

Regarding the profession’s perception, 162 nurses desired more patient time, thereby reflecting concerns about nurse’s expectations. Table 14 illustrates our findings:

In detail, qualitative results regarding self-esteem and perception were analyzed:(i)*Expectation of Nursing and Patient Demands:* Nurses expressed dissatisfaction with the limited time for patient care, thus emphasizing a need for more time to provide quality care and engage in *“conversation at the bedside” (Text 1a, para. 370)*. They also want to listen to patients and meet their holistic demands (Text 1a, para. 39). The most rewarding aspect of their job, patient care, is hindered by time constraints, leading to feelings of guilt and unpaid overtime (Text 1a, para. 129, 435). Time constraints are particularly challenging in dementia patients, where time for discussions with patients or relatives is scarce (Text 3b, para. 139).(ii)*Self-Esteem of their Nursing Role:* Nurses view their role as meaningful, valuable, passionate, and loving (Text 5c, para. 20). Being a nurse is *“doing something meaningful every day” (Text 5c, para. 168).* Joy is derived from helping others, including positive emotions from patients (Text 5c, para. 414). Nurses have a high emotional motivation: Nurses *“put […] heart and soul into [their profession]” (Text 5c, para. 27)*. Nurses view their patients as customers who are central to their role, and, therefore, critiques were made when financial or regulatory constraints limit their ability to meet patient needs. Nurses have altruistic attitudes and want *“to help people, support them during difficult phases of their lives, and alleviate their suffering ” (Text 5c, para. 110)*. Nevertheless, nurses desire to be respected, valued, and recognized (Text 3b, para.219).(iii)*External Perception:* Public media emphasizes negative aspects like *“overwork and shortage” (Text 2b, para. 373)*, thereby contributing to a less attractive image of the profession and deterring potential nursing careers. Additionally, public perception overlooks nurses’ crucial role in patient recovery, which is overshadowed by a focus on doctors (Text 1a, para. 144). *“Not only doctors heal” (Text 1a, para. 440)*. Nurses advocate for a *“respectful treatment” (Text 1a. para 315)* beyond *“1x applause a year” (Text 1a. para 315)* or *“praise and clapping” (Text 5c, para. 304)*. Nurses desire honest *“valuation” (Text 1a, para. 440)*, with one nurse equating nurses’ status with luxury goods (Text 5c, para. 252). Nursing needs societal change, which addresses appropriate collaborations (Text 5c, para. 119) and nurses’ relevance (Text 1a, para. 248). Challenges include interactions with patients’ families, who *“vent frustrations on the nursing staff”* (Text 1a, para. 380), highlighting the need for measures recognizing that *“patients are human beings”* and everyone’s potential need for care (Text 1a, para 397).

#### 4.6.4. Leadership and Line Managers

Leadership influences nurses’ mental and physical health. Line managers and leaders are expected to foster an environment that nurtures resilience and provides essential resources (Text 1a, para. 439). ‘Recognition and Appliance’ was one of the most common subthemes regarding leadership. Table 15 illustrates our findings regarding leadership:

In detail, the qualitative results regarding self-esteem and perception were analyzed:(i)*Recognition and Appreciation:* Nurses emphasized the need for recognition and understanding from leadership. Nurses expressed dissatisfaction with their leaders’ recognition, which reduced their motivation. One nurse summarized: *“It’s crucial for employers to value their employees—this appreciation should be tangible. We wear […] out every day” (Text 1a, para. 282)*. Nurses pay attention to details like correctly writing names and personal anniversaries (Text 4b, paras. 101–102). Another added, *“Employees must be treated with respect and appreciation. Physically demanding work and willingness to step in or double shifts should be valued, not taken for granted” (Text 4b, para. 230)*.(ii)*Codetermination:* Nurses highlighted the necessity to *“include employees in key decisions and value their input” (Text 1a, para. 314)*. They called for communication marked by respect, transparency and honesty, where their concerns are taken seriously (Text 4b, para. 31). Regular staff meetings and employee surveys are vital (Text 2b, para. 505).(iii)*Hierarchies:* Nurses lamented their hospitals’ hierarchies as outdated. *“Physicians often act as ‘gods in white’. Nurses are at the end of the food chain (Text 4b, para. 54).* They prefer flatter structures where their opinions matter (Text 2b, para. 473).

#### 4.6.5. Retention to the Nursing Profession

The previous qualitative assumptions represent the status quo, while the focus is now on retention, the third most common supercategory, visualized by Table 16. Retention can result from prior job satisfaction variables [10,35,37], so we deviated from presenting the most common comments one after the other. Our free-text comments reveal a dichotomy: 177 Nurses cited reasons for considering leaving. In contrast, 131 nurses conveyed factors such as their love for the nursing profession that make them stay (Text 5c, para. 20).

In detail, the qualitative results regarding nurse retention were analyzed:(i)*Reasons to Leave Nursing:* Nurses consider leaving the profession due to multiple factors, including long-term physical strain, health issues, job burnout, and missing opportunities for further training (e.g., becoming a nursing service manager) (Text 6c, para. 128, 309). Nurses’ health is a factor in leaving nursing, as one nurse shared that *“after more than 30 years of demanding work, I am contemplating leaving. Otherwise, I face a future in a wheelchair” (Text 5c, para. 148)*. The strain on nurses is high *“due to fatigue […] or unpredictable calls to step in” (Text 5c, para. 277).* Additionally, the emotional impact of patient deaths is significant, as a nurse mentioned *“witnessing the death of 32 residents in four weeks was overwhelming” (Text 6c, para. 354)*. Furthermore, working conditions matter, as 28 nurses expressed dissatisfaction with aspects like working hours, work–life balance and high workload as reasons they want to quit. This sentiment reflects the challenges and exhaustion faced in the profession. Dissatisfaction leads to the attitude that nurses *“can’t recommend [nursing] to young people” (Text 5c, para. 277).*(ii)*Reasons for Staying in Nursing:* One central reason for staying in the profession is nurses’ deep love for the job, which is visualized by the following quote: *“Nursing is my professional passion. It’s still my dream job, and I can’t imagine doing anything else” (Text 5c, para. 31–33).* Nurses expressed satisfaction with their passion as a reason to stay, thereby described nursing as *“most meaningful and beautiful” (Text 5c, para. 27).* Nursing is perceived as more than just a profession but a vocation worth remaining in (Text 5c, para. 180).

## 5. Discussion

### 5.1. Interpretation of Our Main Findings

Our study aimed to assess self-reported (RQ 1) nurses’ job satisfaction, (RQ 2) the importance of work dimensions, and (RQ 3) retention factors. Our takeaways include:(i)Self-reported Time for patient care is the most essential and dissatisfying variable. ^(RQ 1–2)^(ii)Mostly dissatisfaction in service organization, documentation, co-determination, and payment ^(RQ 1)^ were identified.(iii)Nursing conditions are desired to change according to our findings. ^(RQ 1–2)^(iv)Retention is improvable: 56.8% intend to stay in nursing in the next 12 months. ^(RQ 3)^(v)Health as limit: 31.9% can’t proceed nursing due to their health for 12 months. ^(RQ 3)^(vi)Multifaceted retention factors: age, career choice, opportunities, support people. ^(RQ 3)^

#### 5.1.1. Debate Regarding the Answer of RQ 1: Satisfaction Levels and Status Quo

Regarding nurses’ satisfaction (RQ 1), we revealed *mainly dissatisfaction in*: ‘Time for Patient Care’ (66.7%), ‘Service Organization’ (58.3%), ‘Documentation’ (54.9%), ‘Co-determination’ (52.3%), and ‘Payment’ (52.1%) according to Table 6. Conversely, areas of higher satisfaction concerned ‘Team Cohesion’ (with 21.3% being dissatisfied) and satisfaction with ‘Working Aids’ (with 33.4% being dissatisfied). Our template analysis confirmed insufficient patient time and recommended nursing ratios. Moreover, nurses desired to improve service organization, scheduling, and working hours. Many comments were related to payment, job recognition, leadership appreciation, and health challenges.

Our findings resonate with the literature. Despite various studies [24] and measures since the last decades, we revealed a higher dissatisfaction and higher intention to leave than pre-COVID. We did not investigate other health workers than nurses, but it would be necessary to find out why doctors are more satisfied, as stated by Kramer et al. [8]. From a profession comparison, further implications could be derived. We perceive a lack of time for patient care, thereby highlighting a persistent systemic issue. The NEXT study [24] and other German studies [7,31,32,49] indicate that job satisfaction in nursing can be improved, primarily in time for patients, work-life balance, and workload. Our study also reflects a gratification crisis, which is evidenced by dissatisfaction with payment and appreciation [15,25]. In addition, we confirm leadership’s impact on satisfaction [26,39].

#### 5.1.2. Debate Regarding the Answer of RQ 2: Important Values in Nurses Conditions

Regarding RQ 2, *many variables were over 90% important to nurses*, except for ‘Digitalization’ (68.2%, N = 1740) and ‘Nursing Documentation’ (64.9%, N = 1655). We identified a disparity between importance and satisfaction, thus notably meaning that, e.g., ‘Time for Patient Care’ was most dissatisfying but also viewed by 99.5% (N = 2549) of respondents as essential. Other vital factors include ‘Payment and Salary’ (98.3%, N = 2503), ‘Team cohesion’ (97.7%, N = 2508), ‘Leaders Recognize Suggestions’ (97.3%, N = 2499), and ‘Plannable Working and Rest Times’ (97.3%, N = 2491). In addition, our template analysis revealed a deep commitment to the nursing profession, with frustration arising from inadequate patient time. The prominent theme, ‘Overall Working Conditions & Policies’, highlighted staffing, schedules, remuneration, and appreciation as crucial but under-satisfied factors. Responses stress the importance of social factors, valuation and patient interaction.

Consistent with theories like *Maslow’s Hierarchy of Needs* [63] and *Herzberg’s Two-Factor Theory* [64], our study suggests that nurses prioritize a blend of material and immaterial factors, including salary and social aspects like adequate patient time. This indicates that nurses may subordinate their needs to those of patients, thereby risking self-exploitation and burnout [50]. Even if mental health was not our focus, measures derive from our results. Kramer et al. [8] recognize that mental health problems are common and relate to a ‘helper syndrome’, which can be concluded from many qualitative comments. Our findings generally align with national and international studies that emphasize work demand, salary, leadership, recognition, and work–life balance as key for nurses [26,27,28,29,30,38,39].

#### 5.1.3. Debate Regarding the Answer of RQ 3: Factors to Nurses Retention

The intention to stay in nursing (RQ 3), previously measured in studies like the NEXT study [24], requires updated data reflecting changes such as those brought by COVID-19. Despite adverse pandemic effects, we found a lower intention to leave (10.5%) in our study, compared to 18.4% nearly two decades ago [24]. About 56.8% (N = 1299) of nurses plan to stay in the profession in the next 12 months, but 32.7% are unsure, thereby indicating the potential for improving conditions to retain nurses in their profession.

The multidimensional influence factors regarding retention were analyzed via a *multivariable logistic regression*, with results summarized in Table 17. Older nurses, facing challenges in changing careers, tend to stay, though health issues might prompt a shift. Those living alone are more inclined to remain in nursing, possibly due to less social support, while those in shared households may explore other careers. Our qualitative assumptions promote social aspects (esp. time for patients and reconciling work with family life) and their passion for nursing as retention factors. Additionally, satisfaction with *career choice*, *career prospects*, *payment*, *work/rest times* and *supporting people* significantly impact their decision to stay. Despite contentment with their career choice and finding meaning in their work, nurses are concerned about *payment satisfaction*, thereby indicating an imbalance between their workload and compensation or recognition. Closing this summary, working in *long-term elder care* is negatively associated with retention.

Our study validates that *age*, *satisfactory payment*, and *recognition* significantly contribute to nurses’ retention, with five of the eight key factors in our model closely tied to job satisfaction, thereby highlighting the importance of *favorable working conditions* [48]. Despite some improvements, post-COVID developments still reflect longstanding issues such as *stress*, *heavy workloads,* and *reward imbalances*, as identified in German studies [15,25,32]. Comparing our results to Poland, codetermination regarding work patterns seems important, but in our regression model, it was not included significantly [45]. Internationally, factors like *supportive leadership*, *job autonomy*, *fair compensation,* and *positive work environments* are linked to higher job satisfaction and retention, which is reflected in our work [27,28,29,30,39].

Our research also highlights the *emotional toll*, particularly in long-term care settings like elder care, where challenges include *managing dementia,* and resources are more limited [29]. Nurses often face *emotional distress* and potential burnout from deep patient relationships, thereby leading to considering leaving the profession [7,50].

Additionally, our study reveals crucial factors not extensively covered in previous research, such as *satisfaction with working and rest times* or *career advancement opportunities*, thereby broadening the knowledge spectrum. However, not all influential factors, like leadership, were included in our model due to limited effect sizes. *Effective leadership* is still crucial in nurses’ retention decisions, as evidenced by other studies [10,24,32,49].

### 5.2. Methodical Limitations

Our study, offering a view of nursing conditions and retention, faces limitations.

#### 5.2.1. Generalizability and Sample Focus

Our study’s focus on Bavarian nurses limits its *generalizability* to wider regions or national contexts. The study population mainly consisted of female participants working in inpatient hospital care, with under-representation from other care services. The absence of a comparison group from different areas restricts the scope, thereby highlighting the future need for broader geographic research. The reliance on a *convenience sample* and online survey targeting currently employed nurses may have introduced *self-reporting bias* and missed insights from nurses who have left the profession [56]. Additionally, being conducted *post-COVID-19*, the study might reflect specific pandemic-related stressors, thereby potentially affecting our findings. This approach may not fully represent the diversity of nursing experiences concerning the accuracy and completeness of our results.

#### 5.2.2. Instrument and Survey Limitations

Our survey, developed from validated instruments like the Index for Work Satisfaction [52], the Minnesota Satisfaction Questionnaire [53], the Work-Life Balance Scale [54] and national nursing studies [12,15,43,55], aimed to cover a wide range of personal, organizational, job-related, and demographic factors. We aimed to cover a *broad spectrum of themes to ensure a well-rounded analysis*. In our survey, we included further literature-based variables, primarily ‘Team Cohesion,’ ‘Corporate policies, ‘ and aspects of ‘Leadership’. Additionally, existing surveys often come with licensing restrictions, thereby limiting their use. To overcome this and contribute to the academic community, we developed our survey to be *open-access* available. Despite a pretest confirming the survey’s relevance, the survey’s depth and detail were limited. We utilized a modified Likert scale, *excluding a neutral option*, to encourage decisive responses. Furthermore, our survey did not employ an index, because it did not incorporate all variables from the original validated questionnaires.

Our study intentionally delivers a big picture of nurses’ conditions but needs deeper insights. For example, Kilanska et al. [45] examined *work patterns*, including shifts and autonomy to influence working times and connection to retention, which needs to be recognized in further research. In addition, our categorization of *work areas* could have been more granular, thereby distinguishing between intensive care and general wards. This should be more detailed in further iterations, as specific areas like intensive care may face higher stress levels and retention challenges [65]. Furthermore, our study did not measure nurses’ *personality traits*, but previous studies highlighted a effect of personalities [66]. We did not measure nurses’ *mental health*, even though this will be more common regarding Kramer et al. [8,67]. Analyzing mental health and nursing retention dependencies would be interesting. Regarding our survey, future governmental-supported research should include the *development of a more detailed and valid national comprehensive survey*.

#### 5.2.3. Data Analysis

Our *cross-sectional* study employed open-ended and close-ended questions, thus leading to statistical and template analysis. The *quantitative analysis* focused on close-ended questions, which provide relatively limited information about the assessed variables. Additionally, relying on the Likert point scale for some dimensions introduces subjectivity and potential response bias. The regression analysis, identifying predictors of job retention, showed a Nagelkerke R^2^ value of 0.38. While considered good by Backhaus et al. [58], other influences might not be included in our model. The *template analysis* of the open-ended questions provided additional insights and explored each dimension more thoroughly. Nevertheless, our analysis was self-reported, and despite a quality assurance with researchers checking vice versa through their coding, response bias could have affected our findings.

### 5.3. Recommendations and Implications for Nursing Practice, Managers, and Policymakers

Despite our limitations, employers should prioritize improving dimensions that contribute to nurse satisfaction and retention. Derived from our results, we provide the following ideas to enhance nurses’ conditions (including satisfaction and retention):(i)Patient Time: Digitization, recruiting, process optimization, and debureaucratization.(ii)Career: Establish trainings, career development, and start image campaigns.(iii)Full-Time Work: Enable and promote full-time work and respect life situations.(iv)Shift Reform: Reliable shifts, maximum consecutive shifts, and mandatory downtime.(v)Staffing: Allocate staff for administration, thus freeing nurses for patient care.(vi)Compensation: Re-evaluate pay structures to be fair and performance-oriented.(vii)Resilience: Develop health management focusing on physical and mental resilience.

We recommend that policymakers and managers (i) ensure more time for direct patient care because this is one core value of nurses, which is actually dissatisfying regarding our results. More patient time might be achieved with digitally optimized processes and additionally supporting staff. Providing more time also includes de-bureaucratization, as documentation was described as too time-consuming. Nevertheless, it is essential to establish a better image with (ii) image campaigns and improved working conditions, including career development options. Derived from our regression analysis, which showed that more working hours facilitate the stay in nursing, this leads to the implication to (iii) promote full-time work. Yet, to enable a work-life balance, mother shifts and an (iv) shift work reform, with less unplanned stepping in, a maximum consecutive shift duration, and mandatory reliable downtime, need to be introduced. To facilitate supportive environments for nurses, we also recommend recruiting (v) additional staff for administrative tasks, thereby freeing nurses for direct patient care. We recommend (vi) re-evaluating compensation, as there seems to be a gratification crisis. Health appears to be limiting to proceeding nursing, so health management systems should be implemented.

## 6. Conclusions

Amid escalating demands and staff shortages in Germany’s healthcare, comprehending nurses’ expectations, satisfaction, and retention factors is vital. Our study addresses a national research gap, thereby shedding light on potential starting points for improvements.

### 6.1. Job Satisfaction and Influence on Retention

The primary conclusion, summarized in Table 18, is that several nursing job conditions require improvement, particularly in providing adequate patient care time, which was a factor with which more than 6.7% (N = 1533) of nurses expressed dissatisfaction.

Our template analysis reinforces that sufficient patient time is the most important and primarily dissatisfying factor for nurses, who are often socially oriented and enjoy helping others. Furthermore, over 58.3% (N = 1340) of nurses responded as being dissatisfied with the service organization, thus indicating a need for more reliable schedules and fewer unplanned shifts. These findings align with existing research, thereby highlighting the necessity to address nurses’ working conditions while also recognizing the profession as a valued, sacrificial, and beloved vocation [26,27,28,29,38,39,50].

More than half of the nurses were dissatisfied with *’Documentation’*, *’Co-Determination Rights’*, and *’Payment’*, thereby highlighting key areas for action. Moreover, a discrepancy exists according to Table 6 between *importance* and *actual satisfaction* in job dimensions, with the highest contrast in ‘Time for Patient Care’, ‘Service Organization,’ and ‘Payment.’ Our recommendations to improve *nursing satisfaction and retention* (Section 5.3) take that into account, as expectations and satisfaction are often divided.

Our conclusion also includes factors influencing nurses’ *intention to stay* in the profession. Older nurses are less likely to leave, thus suggesting a focus on retention strategies for *younger nurses*. Living conditions, living with partners, working in elderly care, and shorter working hours appear to impact the intention to stay negatively. Satisfaction in *career choice* and *training opportunities* significantly boosts retention prospects. Besides ‘Payment and Salary’, predictable *’Work and Rest Schedules’* are crucial for retaining nurses. Our findings on the adverse effects of supporting people in tough situations indicate a need for further research to explore how sustained altruistic efforts might impact *nurse retention*.

### 6.2. Further Research

Our study identifies improvable workplace conditions, notably in *patient care time*, *service organization*, *documentation*, *codetermination rights,* and *payment*. In addition to evidence-based measures, we encourage other researchers to study the interventions needed in our identified key areas. We recommend establishing regular, comprehensive national surveys in healthcare facilities to assess nursing conditions through a *longitudinal analysis*.

The scientific community will benefit from creating a *nursing panel*, thereby providing deeper insights and evaluating the impacts of pandemics or policy interventions. Moreover, we urge the further development of our survey, including broader populations, comparison between nursing units, and evolving the impact of *digitalization and robotics* in nursing.

## Figures and Tables

**Table 1 healthcare-12-00172-t001:** International evidence of influence factors with respect to nurses’ satisfaction and retention.

Characteristics	Influence Factor	Effect	Reference (s)
	Good Health and Physiological Status	+ ^1^	[29]
**Personal** *	Higher Educational Status	+ ^1^	[38]
	Marital Status and Having Children	+ ^1^	[1,38]
	Older (>30 Years) and Having Job Experience	+ ^1,2^	[29,38,41,43]
**Job** *	Leadership Support, Supervision, and Authenticity	+ ^1^	[26,27,29,38,39,42]
	Leadership Style: Passive-Avoidant, Laissez-Faire	− ^1^	[27,39]
	Leaders Themselves Unsatisfied	− ^1^	[27]
	High Job Stress and Workload	− ^1,2^	[1,20,27,29,40]
	Professional Growth and Development	+ ^1^	[1,20,38,39]
	Job Autonomy and Freedom	+ ^1^	[1,27,28,29,45]
**Organizational** *	Financial Factors (Salary and Benefits)	+ ^1,2^	[42]
	Staff Relationship (e.g., Social Support)	+ ^1^	[1,27,29,38]
	Patient Relationship	+ ^1^	[29]
	Involvement and Codetermination	+ ^1^	[20,27,29,45]
	Urban Employer Location	− ^1^	[28]
	Physical Working Environment and Equipment	+ ^1^	[28]

^1^ Effect on satisfaction; ^2^ Effect on retention; * Factors categorized by Penconek et al. [27] and Aloisio et al. [29].

**Table 2 healthcare-12-00172-t002:** Significant and literature-based IVs, included in regression analysis, with scale level.

Influence Factor	Scale	Influence Factor	Scale
Career and Training Opportunities ^1,+^	Ordinal	Payment and Salary ^2,+^	Ordinal
Plannable Working and Rest Times ^2,+^	Ordinal	Support People in Life Situations ^2,+^	Ordinal
Working Hours ^3^	Ordinal	Age ^1^	Ordinal
Illegitimate Partners, Living Apt. ^1^	Binary	Work Area: Stationary Elderly Care ^2^	Binary
Career Choice: Entering Again ^1^	Binary	Career Choice: Maybe, If It Changes ^1^	Binary

^1^ Personal characteristics; ^2^ Job characteristics; ^3^ Organizational characteristics; ^+^ Satisfaction.

**Table 3 healthcare-12-00172-t003:** General characteristics of employer and organizational policies.

Variables	Relevance/ Importance				Satisfaction			
	+ +	+	−	−−	+ +	+	−	−−
Career and Training Opportunities	35.1% (898)	52.3% (1339)	11.6% (296)	1.1% (28)	21.0% (483)	44.2% (1016)	25.8% (592)	9.0% (206)
Payment and Salary	66.0% (1681)	32.3% (822)	1.7% (43)	0.0% (1)	7.9% (184)	40.0% (926)	38.8% (889)	13.3% (307)
Codetermination Right(s)	38.7% (992)	53.1% (1361)	7.7% (198)	0.4% (10)	10.9% (250)	36.8% (843)	38.5% (881)	13.8% (317)
Work Promotes Health	66.4% (1707)	28.4% (731)	4.9% (126)	0.2% (6)	10.7% (246)	45.6% (1045)	33.5% (768)	10.2% (233)
Work–Family Reconciliation	78.1% (2006)	18.9% (485)	2.5% (65)	0.4% (11)	11.8% (271)	38.7% (889)	34.7% (797)	14.9% (342)
Individual Working Hours	56.9% (1458)	34.2% (877)	7.8% (201)	1.1% (27)	15.8% (364)	42.1% (971)	31.9% (735)	10.2% (236)
Leaders Recognize Suggestions	58.7% (1507)	38.6% (992)	2.6% (66)	0.2% (4)	16.8% (384)	42.3% (966)	28.8% (659)	12.1% (276)

^++^ Very important|fully satisfied; ^+^ Important|rather satisfied; ^−^ Less important|rather dissatisfied; ^−−^ Irrelevant|not at all satisfied.

**Table 4 healthcare-12-00172-t004:** General characteristics of the nursing and care organization.

Variables	Relevance/ Importance				Satisfaction			
	+ +	+	−	−−	+ +	+	−	−−
Reliable Service Organization	63.4% (1629)	31.8% (817)	4.5% (115)	0.3% (7)	8.4% (192)	33.4% (767)	36.9% (848)	21.4% (492)
Plannable Working and Rest Times	67.2% (1721)	30.1% (770)	2.5% (63)	0.2% (6)	11.1% (256)	39.8% (914)	37.1% (853)	12.0% (275)
Time for Patient Care	83.9% (2151)	15.5% (398)	0.5% (13)	0.0% (1)	8.4% (197)	24.8% (570)	36.7% (844)	30.0% (689)
Nursing Documentation	18.2% (464)	46.7% (1191)	32.6% (831)	2.6% (66)	6.8% (154)	38.4% (875)	40.0% (912)	14.9% (340)
Working and Auxiliary Tools	53.4% (1369)	42.5% (1089)	4.0% (103)	0.1% (3)	11.2% (258)	55.3% (1269)	27.3% (627)	6.1% (141)
Digitalization	24.8% (633)	43.4% (1107)	26.8% (683)	5.0% (127)	11.0% (245)	40.1% (892)	33.1% (738)	15.8% (352)

^++^ Very important|fully satisfied; ^+^ Important|rather satisfied; ^−^ Less important|rather dissatisfied; ^−−^ Irrelevant|not at all satisfied.

**Table 5 healthcare-12-00172-t005:** General characteristics of social aspects.

Variables	Relevance/Importance				Satisfaction			
	+ +	+	−	−−	+ +	+	−	−−
Team Cohesion (e.g., Relationship to Colleagues)	73.3% (1881)	24.4% (627)	2.2% (56)	0.1% (2)	28.9% (667)	49.8% (1147)	17.3% (398)	4.0% (92)
Relationship with Managers	40.2% (1030)	49.6% (1270)	9.6% (246)	0.7% (17)	17.1% (393)	42.9% (984)	29.3% (672)	10.7% (246)
Support People in Tough Situations	60.8% (1557)	35.2% (901)	3.7% (95)	0.3% (8)	13.7% (306)	48.5% (1086)	31.4% (703)	6.4% (144)

^++^ Very important|fully satisfied; ^+^ Important|rather satisfied; ^−^ Less important|rather dissatisfied; ^−−^ Irrelevant|not at all satisfied.

**Table 6 healthcare-12-00172-t006:** Summary of job and organizational characteristics.

Top 5: Variables	Important ^1^	Top 5: Variables	Dissatisfied ^2^
Time for Patient Care	99.5% (2549)	Time for Patient Care	66.7% (1533)
Payment and Salary	98.3% (2503)	Reliable Service Organization	58.3% (1340)
Team Cohesion	97.7% (2508)	Nursing Documentation	54.9% (1252)
Leader Recognize Suggestions	97.3% (2499)	Co-Determination Right(s)	52.3% (1198)
Plannable Working and Rest Time	97.3% (2491)	Payment and Salary	52.1% (1196)

^1^ Very important and important; ^2^ Rather dissatisfied and not at all satisfied.

**Table 7 healthcare-12-00172-t007:** Retrospective view on the occupational decision; N = 2292.

Variable	Category	Count (%)
Would you enter nursing a second time?	Yes, again and again	27.6%
	Maybe, when the conditions change	55.3%
	No, never again (I regret)	17.1%

**Table 8 healthcare-12-00172-t008:** Intention to stay in nursing over the next 12 months; N = 2286.

Variable	Category	Count (%)
Do you plan to stay faithful to the nursing profession in the next 12 months?	Yes, I plan to stay	56.8%
	Maybe, when conditions change	32.7%
	No, I will not stay	10.5%

**Table 9 healthcare-12-00172-t009:** Health-related feasibility of staying in nursing over the next 12 months; N = 2036.

Variable	Category	Count (%)
Is staying in nursing feasible in terms of your health in the next 12 months?	Yes, it is possible due to good health	14.7%
	Maybe, when conditions change	53.3%
	No, it is not possible due to my health	31.9%

**Table 10 healthcare-12-00172-t010:** Model visualization: significant factors contributing to employee retention, with N = 1702.

Variable (N = 1702), R^2^ = 0.38	B	S.E.	Wald	Sig.	OR	95% CI
Age	0.21	0.05	20.85	<0.000	1.23	1.13–1.35
Illegitimate Partners, Living Apt.	0.95	0.32	8.60	0.003	2.59	1.37–4.89
Work Area: Stationary Elder Care	−0.34	0.15	5.58	0.018	0.71	0.53–0.94
Working Hours per Week	0.18	0.09	4.38	0.036	1.20	1.01–1.42
Career Choice: Yes, entering again nursing	3.15	0.22	197.04	<0.000	23.32	15.02–36.20
Career Choice: Maybe, if condition change	1.09	0.16	46.98	<0.000	2.98	2.18–4.07
Career and Further Training Opportunities ^+^	0.28	0.07	14.61	<0.000	1.33	1.15–1.54
Payment and Salary ^+^	0.12	0.06	3.92	0.05	1.23	1.00–1.27
Working and Rest Times ^+^	0.24	0.09	7.80	0.01	1.28	1.08–1.51
Supporting People in Tough Life Situations ^+^	−0.33	0.09	14.31	<0.000	0.72	0.61–0.85

^+^ Satisfaction; R^2^ Nagelkerkes determination; B Regression coefficient; S.E. Stand. error; CI Confidence interval.

**Table 11 healthcare-12-00172-t011:** Template analysis: (Sub-) themes sorted in descending order, with N = 2202.

Themes	Subthemes	Count (%)	Count (%)
Overall Working Conditions and Policies	Staffing & Nursing Ratios (i)	13.94%	44.87%
	Duty Scheduling and Working Hours (ii)	9.40%	
	Payment and Compensation (iii)	9.31%	
	Health-Related Challenges (iv)	6.03%	
	Debureaucratization (v)	3.36%	
	Corporate and Team Culture (vi)	2.81%	
Regulatory and Given Framework Conditions	Digitalization (i)	7.26%	17.48%
	Regulatory Specifications and Inspections (ii)	5.54%	
	Training and Education (iii)	3.04%	
	Working Aids and Equipment (iv)	1.63%	
Retention to the Nursing Profession	Reasons to Leave Nursing (i)	9.31%	15.26%
	Reasons Staying in Nursing (ii)	5.95%	
Self-Esteem and Nursing Profession	Expectation of Nursing and Patient Demands (i)	7.36%	12.44%
	Self-Esteem of the Own Role (ii)	3.00%	
	External Perception of Nurses (iii)	2.09%	
Leadership and Line Managers	Recognition and Appreciation (i)	8.17%	9.95%
	Codetermination (ii)	1.32%	
	Hierarchies (iii)	0.45%	

**Table 12 healthcare-12-00172-t012:** Summary of the qualitative assessment of overall working conditions and policies.

	Subthemes	Quote Visualizing the Status Quo and Intervention Potential
i	Staffing Ratios	“Distributing the workload across more shoulders” (Text 2b, para. 267).
ii	Scheduling *	“No more than ten days [of consecutive service]” (Text 2b, para. 91).
		“Nursing needs a [reliable] downtime” (Text 3b, para. 124).
iii	Payment *	“Payment doesn’t reflect mental and physical effort” (Text 5c, para. 68).
		“Better compensation for [extra] and holiday [work]” (Text 2b, para. 523).
iv	Health	“Need for more back-friendly work […] and support” (Text 7c, para. 370).
		“Rising aggression among patients is alarming” (Text 5c, para. 291).
		“Stress caused by under-staffing and time pressure” (Text 4b, para. 154).
v	Bureaucracy	“Time-consuming paperwork should be delegated” (Text 1a, para. 166).
vi	Culture	“Service meetings should occur at least once a month” (Text 4b, para. 200).

Most qualitative data regarding (i), (ii), and (iii); * Qualitative effects in alignment with regression analysis.

**Table 13 healthcare-12-00172-t013:** Summary of the qualitative assessment of regulatory and framework conditions.

	Subthemes	Quote Visualizing the Status Quo and Intervention Potential
i	Digitalization	“Digitalization must become easier, more automated” (Text 3b, para. 117).
		“We need to overcome interface problems” (Text 1a, para. 310).
ii	Specifications	“Nurses should be able to retire at age 63” (Text 1a, para. 109).
		“Healthcare is a state duty” (Text 1a, para. 180).
iii	Training *	“Degrees leading up to physician assistant or physician” (Text 5c, para. 56).
		“Nursing academia should be more appealing” (Text 1a, para. 545).
		“Nursing trainees should receive superior training” (Text 1a, para. 528).
		“[Facilitate] language skills, [with] courses and exams” (Text 2b, para. 390).
iv	Equipment	“Better provision of nursing aids is necessary” (Text 1a, para. 434).
		“Larger patient rooms that facilitate […] mobilization” (Text 3b, para. 71).
		“Separate rooms for breaks and administration” (Text 3b, para. 382).

Most qualitative data regarding (i) and (ii); * Qualitative effects in alignment with regression analysis.

**Table 14 healthcare-12-00172-t014:** Summary of the qualitative assessment of self-esteem and nursing perception.

	Subthemes	Quote Visualizing the Status Quo and Intervention Potential
i	Patient Demands *	“More time for the individuals in our care” (Text 4b, para. 92).
ii	Self-Esteem *	“I love my job and put my heart and soul into it” (Text 5c, para. 27).
		“To put a smile on people’s faces […] is priceless” (Text 5c, para. 27).
		“I wanted to help people […] and alleviate” (Text 5c, para. 110).
		“This job is doing […] meaningful every day” (Text 5c, para. 168).
iii	External Perception	“The depiction of nursing shouldn’t be negative” (Text 2b, para. 373).
		“Valuation [and] appreciation […] are important” (Text 1a, para. 440).

Most qualitative data regarding (i) and (ii); * Qualitative effects in alignment with regression analysis.

**Table 15 healthcare-12-00172-t015:** Summary of the qualitative assessment of leadership and line managers.

	Subthemes	Quote Visualizing the Status Quo and Intervention Potential
i	Recognition	“It’s crucial for employers to value [nurses]” (Text 1a, para. 282).
ii	Codetermination	“Include employees in key decisions, value input” (Text 1a, para. 314).
iii	Hierarchies	“Very hierarchical structures are common in clinics” (Text 4b, para. 54).
		“It is time for parity with the nursing staff” (Text 4b, para. 54).

Most qualitative data regarding (i) and (ii).

**Table 16 healthcare-12-00172-t016:** Summary of the qualitative assessment of factors leading to retention (intention to stay).

	Subthemes	Quote Visualizing the Status Quo and Intervention Potential
i	Leaving	”[Because] demanding physical & shift work, I am leaving” (Text 5c, para. 148).
		“Unsure if I can handle psychological stress” (Text 6c, para. 354).
		“Private constraints, unfavorable working hours *” (Text 5c, para. 118).
		“Fatigue after work or unpredictable calls to step in *” (Text 5c, para. 277).
		“[Nursing] doesn’t allow […] a social environment” (Text 5c, para. 277).
ii	Staying	“Nursing is my […] passion. It’s still my dream job” (Text 5c, para. 31).
		“Nurses consider [nursing as] meaningful and beautiful” (Text 5c, para. 10).

Most qualitative data regarding (i); * Qualitative effects in alignment with regression analysis.

**Table 17 healthcare-12-00172-t017:** Model summary: factors influencing nurses’ retention.

Likely to Stay in Nursing	Likely to Leave Nursing
Age (being an Older Nurse)	Working in Stationary Elderly Care
Living Apart/Alone	Satisfied with Supporting People
Satisfied with Career Choice and Opportunities	in Tough Situations
Satisfied with Payment and Salary	
Satisfied with Working and Rest Times	

**Table 18 healthcare-12-00172-t018:** Job satisfaction and factors influencing self-reported retention (regression analysis).

More Than 50% Dissatisfied	Factors Influencing Retention
Sufficient Time for Patient Care ^1^	Demographics (Age ^1^, Living Conditions)
Reliable Service Organization ^1^	Working Area and Working Hours
Nursing Documentation	Career Choice ^2^ and Satisfaction with Career Opportunities ^2^
Codetermination Rights	Satisfaction with Payment and Salary
Payment and Salary ^1^	Satisfaction with Working and Rest Times ^1^
	Satisfaction with Supporting People ^2^

^1^ Deviation between expectation and satisfaction more than 50%. ^2^ Highly significant factors.

## Data Availability

Our research data are available from the first author Domenic Sommer (domenic.sommer@th-deg.de) and will be provided as individual material upon request due to privacy/ethical restrictions.

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
