# Peer review of "Nurses’ Workplace Perceptions in Southern Germany—Job Satisfaction and Self-Intended Retention towards Nursing"

_healthcare, 2024, doi:10.3390/healthcare12020172_

Round 1

Reviewer 1 Report (Previous Reviewer 1)

Comments and Suggestions for Authors

Author Response

(Please see the attachment)

Dear Reviewer, We highly appreciate your insightful feedback. We've diligently revised our manuscript, addressing each point to enhance its quality and relevance. The improvements are detailed in the attached PDF.

Your guidance has been crucial in improving our work, and we are eager to contribute meaningfully to nursing research. We appreciate the opportunity to refine our paper and look forward to any further suggestions. Thank you again for your valuable input.

Sincerely, The Corresponding Author

Reviewer 2 Report (New Reviewer)

Comments and Suggestions for Authors

The subject of this study deals with important practical issues in the field of nursing.

However, it seems that a major revision is needed to increase the readability of the thesis.

It seems that it should be organized according to the framework of the research paper writing, focusing on the research topic.

You attempted a mixed research methodology in which qualitative and quantitative studies were conducted together, but it is necessary to present reliable evidence when analyzing the results of mixed studies. The reliability of the research tools used in quantitative research should also be presented. 

No reference was found to IRB approval for ethical consideration of the subjects.

If you go into the details, ã…‘t is necessary to present the necessity of research by summarizing and describing previous studies related to the research topic in the introduction. In other words, please review the  previous studies related to  nurse's job satisfaction and self-Intended retention and present it in a summary and present the necessity of this study.

Since there is a format of thesis  presented in the journal, I recommend you delete 1.3. Structure of the Paper.

Why don't you summarize the contents of  2. Related Work and include it in the introduction. The number of samples in this study is quite large, but it is limited to one region. I don't think there is a need to expand from related work to national and global perspectives.

It would be nice to reduce the number of tables by organizing the research results table around the core contents.

Author Response

(Please see the attachment)

Dear Reviewer, We highly appreciate your insightful feedback. We've diligently revised our manuscript, addressing each point to enhance its quality and relevance. The improvements are detailed in the attached PDF. Your guidance has been crucial in improving our work, and we are eager to contribute meaningfully to nursing research. We appreciate the opportunity to refine our paper and look forward to any further suggestions. Thank you again for your valuable input.

Sincerely, The Corresponding Author

Reviewer 3 Report (New Reviewer)

Comments and Suggestions for Authors

This is an interesting and necessary study. In the face of nursing staff shortages, understanding job satisfaction and retention is critical. In addition, it has been carried out with a large sample of nurses (N=2572). However, it has some limitations that need to be reflected in the discussion section:

       No validated scale was used to measure whether the nurses surveyed had mental health problems such as anxiety, depression, post-traumatic stress disorder, or burnout. These problems occur frequently among nurses, and it would be interesting to detect them and see if they are related to the variables in this study. In the study by Kramer et al. (2021) conducted with German healthcare workers, nurses reported, in principle, higher ratings on all questions of subjective burden and stress than doctors and other hospital staff. Moreover, these are problems that already existed before the pandemic. In Maharaj et al.'s (2018) study of Australian nurses, prevalence rates of depression, anxiety, and stress were found to be 32.4%, 41.2%, and 41.2%, respectively.

Kramer, V., Papazova, I., Thoma, A., Kunz, M., Falkai, P., Schneider-Axmann, T., Hierundar, A., Wagner, E., & Hasan, A. (2021). Subjective burden and perspectives of German healthcare workers during the COVID-19 pandemic. European archives of psychiatry and clinical neuroscience271(2), 271–281. https://doi.org/10.1007/s00406-020-01183-2

Maharaj, S., Lees, T., & Lal, S. (2018). Prevalence and Risk Factors of Depression, Anxiety, and Stress in a Cohort of Australian Nurses. International journal of environmental research and public health16(1), 61. https://doi.org/10.3390/ijerph16010061

       As for the area of work, only the following was taken into account: Stationary Elderly Care. Other areas of work of the nurses surveyed were not considered. This is also a limitation of the study. In some areas such as the ICU (Hung et al., 2022), nurses experience more stressful workplace conditions, making them more vulnerable to high levels of depression compared with those working in other healthcare settings. This may cause nurses in these units to leave the profession.

Huang, H., Xia, Y., Zeng, X., & Lü, A. (2022). Prevalence of depression and depressive symptoms among intensive care nurses: A meta-analysis. Nursing in critical care27(6), 739–746. https://doi.org/10.1111/nicc.12734

       The risk of abandonment of work also depends on work shifts (KilaÅ„ska et al., 2019) and has not been considered in quantitative analysis.

KilaÅ„ska, D., Gaworska-KrzemiÅ„ska, A., Karolczak, A., Szynkiewicz, P., & Greber, M. (2019). Work patterns and a tendency among Polish nurses to leave their job. Medycyna pracy70(2), 145–153. https://doi.org/10.13075/mp.5893.00727

       Indicate in the Methodology section the ethical considerations of the study. Add the details of the Human Research Ethics Committee.

       The following tables of results should be presented, showing the different percentages according to gender instead of the total percentages.

Table 3. General Characteristics of Employer and Organizational Policies.

Table 4. General Characteristics of the Nursing and Care Organization

Table 5. General Characteristics of Social Aspects

Table 6. Summary of Job & Organizational Characteristics

Author Response

(Please see the attachment)

Dear Reviewer, We highly appreciate your insightful feedback. We've diligently revised our manuscript, addressing each point to enhance its quality and relevance. The improvements are detailed in the attached PDF. Your guidance has been crucial in improving our work, and we are eager to contribute meaningfully to nursing research. We appreciate the opportunity to refine our paper and look forward to any further suggestions. Thank you again for your valuable input.

Sincerely, The Corresponding Author

Reviewer 4 Report (New Reviewer)

Comments and Suggestions for Authors

Dear Authors,

I trust this message finds you well. I've thoroughly reviewed your article, and some specific recommendations could further enhance its overall quality and impact. I offer constructive suggestions to refine various paper sections with that in mind.

My recommendations are intended to build upon the solid foundation you've established. Your research is undoubtedly valuable, and these proposed improvements ensure its significance is effectively communicated to your readers.

In the abstract several aspects can be improved scientifically to enhance its clarity and informativeness:

  • Clarity of Research Objectives: The abstract starts with a general statement about the importance of understanding job satisfaction and retention in nursing. It would be more scientifically precise to explicitly state the research objectives or hypotheses the study aimed to address. What specific aspects of job satisfaction and retention were investigated?
  • Methodology Overview: The abstract could benefit from a brief overview of the methodology used in the study. Mentioning the study design (cross-sectional), sample size (2,572 nurses), and data collection methods would give readers a clearer understanding of how the research was conducted.
  • Key Findings: While the abstract mentions the top areas identified for enhancement, such as patient care time and nursing documentation, it could provide more specific findings or statistics related to these areas. For example, what percentage of nurses expressed dissatisfaction with patient care time? What were the main issues with nursing documentation?
  • Regression Analysis: When mentioning the multivariable regression analysis, specify some key factors influencing nurses' intentions to stay in the profession. This could include significant demographic factors, working conditions, or job satisfaction-related variables. A glimpse of these findings would make the abstract more informative.
  • Limitations and Potential Bias: It's mentioned that there are limitations and potential biases due to convenience sampling. Expand on these limitations briefly by specifying what they are and how they might affect the study's results. This helps readers assess the reliability of the findings.
  • Future Research Directions: The abstract briefly mentions that future research should broaden its scope to investigate nurse job satisfaction and retention. Suggesting specific areas or aspects that future research could explore would be helpful. For instance, how might digitization impact job satisfaction and retention in nursing?
  • Overall Conclusion: The abstract should conclude with a clear and concise statement of the main implications or takeaways from the study. Based on the findings, what are the practical implications for nursing practice or policy?

 Introduction: Here are some points to improve the scientific quality of this section:

  • Clarity and Relevance: The introduction should start by clearly stating the research problem or question. In this case, it's about the incidence of anxiety and depression among health workers during the COVID-19 pandemic in Latvia. The introduction should directly address this topic to provide context for the study.
  • Specificity of the Research Problem: While the ageing population and healthcare challenges are mentioned, it would be more scientifically precise to clearly articulate the specific research problem related to the mental health of health workers in Latvia during the pandemic. What is the exact issue or gap in knowledge that this study addresses?
  • Citation of Relevant Studies: When discussing challenges in nursing and the impact of the COVID-19 pandemic, it's important to cite specific studies or sources that support these claims. This adds credibility to the introduction and helps readers understand the background of the research.
  • Research Gap and Objectives: The section mentions that the study addresses a research gap in understanding nursing conditions and job satisfaction. It's essential to explicitly state what this research gap is and how this study intends to fill it. Also, clearly outline the research objectives or hypotheses the study aims to investigate.
  • Contribution of the Study: While the section mentions the study’s contributions, providing more detail would be helpful. For instance, how does this study's approach or scope differ from existing research? What is unique about the data collection methods or the specific focus on Latvian health workers during the pandemic?
  • Structure of the Paper: The section outlines the structure of the paper, which is good practice. However, it can be more concise. Instead of describing what each section will contain, you can briefly list the main sections (e.g., Literature Review, Methodology, Results, Discussion, Conclusion) and mention any supplementary sections as needed.
  • Clear Transition to the Next Section: Ensure the transition to the next section is smooth. In this case, the transition to the literature review should be clear, indicating how the review of existing literature relates to the research problem and objectives.

The "Related Work" section provides a comprehensive overview of existing research on job satisfaction and retention among nurses, both from a national and international perspective. Here are some points to further improve it:

  • Clarity and Organization: While the section contains valuable information, it can be challenging to follow due to its length and complexity. Consider breaking it down into subsections or using bullet points to highlight key factors and findings for better readability.
  • Citations and References: Ensure that each piece of information or data presented is correctly cited from reputable sources. This enhances the credibility of the information and allows readers to access the original studies for more in-depth information.
  • Focus on Relevance: While providing a broad overview is essential, emphasize the aspects of the existing research most relevant to your study in Latvia during the COVID-19 pandemic. Highlight findings or factors that have direct implications for your research objectives.
  • Research Gap Clarification: Clearly state how the existing research falls short or needs to address your study’s specific context or research questions fully. Identify the research gap that your study aims to fill in this context.
  • Alignment with Research Objectives: Ensure that the information presented in this section directly connects to your study's objectives and hypotheses. Readers should understand why this background information is crucial to your research.
  • Recent Sources:  consider including more recent studies or findings directly related to nursing job satisfaction and retention nationally and internationally.
  • Synthesis and Comparison: After presenting a variety of factors and findings, consider providing a brief synthesis or comparison of the key factors that influence job satisfaction and retention across different contexts (national and international). What common themes or differences emerge from the literature?
  • Concluding Transition: End the "Related Work" section with a clear transition to the next section of your paper. Indicate how the literature review sets the stage for your research and hypotheses.

The methodology is quite detailed and well-structured. However, there are a few suggestions to improve the scientific rigour and clarity:

  • Sampling Method and Representativeness: While you mention using a convenience sampling strategy due to the unavailability of a random dataset, it's essential to acknowledge this sampling method’s limitations explicitly. Discuss the potential biases that may arise from this sampling approach and how they might impact the generalizability of your findings to the entire population of nurses in Bavaria.
  • Data Collection Form: Provide more details about the specific questions asked in your survey, especially the open-ended questions. This will help understand the depth and breadth of the information collected from participants.
  • Data Preparation and Cleaning: Explain in more detail how you handled missing data and what criteria you used for removing incomplete responses. This transparency is essential for readers to assess the robustness of your analysis.
  • Multivariate Model Diagnostics: In the section about multivariate model diagnostics, consider providing more information about the specific tests or procedures you used to assess model assumptions and the goodness of fit. Discuss any potential limitations or assumptions associated with your regression model.
  • Template Analysis of Open-Ended Questions: While describing the steps involved in template analysis, consider summarising the key themes or findings that emerged from the qualitative analysis. This will give readers a sense of the qualitative insights gained from the open-ended responses.
  • Data Protection and Ethical Considerations: Mention any steps to ensure participant confidentiality and ethical considerations related to data collection and analysis. This could include obtaining informed consent, data anonymization, and compliance with data protection regulations.
  • Robustness and Validity: Discuss the steps taken to ensure the robustness and validity of your survey instrument, especially since you adapted existing instruments. Mention any pilot testing or validation procedures to ensure the questions accurately measure the constructs of interest.
  • Appendix Inclusion: Consider including an appendix with the complete survey questionnaire or a representative sample of survey questions to provide readers with a more comprehensive understanding of the data collected.

Results:  there are several areas where improvement can be done:

  • Clarity and Structure: The section is long and could benefit from improved clarity and structure. Consider breaking it down into smaller subsections with clear headings. This will make it easier for readers to navigate and understand the content.
  • Geographical Clarification: The description of the study population mentions that the study includes nurses mainly from Bavaria, but the article's title refers to Latvia. This discrepancy needs to be addressed and clarified. If the survey primarily involves Bavarian nurses, the title and abstract should accurately reflect this.
  • Statistical Significance: When presenting demographic information, it's essential to highlight statistically significant differences or relationships. This can help readers understand which demographic factors may significantly impact the study's outcomes.
  • Clarity in Satisfaction Analysis: Clearly defining what each variable represents is crucial in the satisfaction with employer and organizational policies section. Please briefly explain why these variables were chosen and how they relate to job satisfaction.
  • Interpretation of Results: While the section presents percentages and numbers related to satisfaction and importance, it lacks an in-depth interpretation of the findings. What do these disparities between importance and satisfaction levels indicate? Are there any implications for healthcare worker well-being or retention? These aspects need to be discussed more comprehensively.
  • Comparison and Benchmarking: To enhance the scientific rigour of the study, consider comparing your findings with relevant literature or benchmarking against similar studies. This can provide context for the results and help readers understand whether the levels of satisfaction and importance are typical or exceptional.
  • Regression Model Explanation: In the section describing the logistic regression model, provide a more detailed explanation of the variables included, their selection criteria, and why they were chosen. Additionally, discuss the rationale for excluding certain variables that might have been relevant. It's also essential to mention any potential limitations of the model, such as multicollinearity or assumptions.
  • Discussion of Practical Implications: At the end of this section, consider discussing the practical implications of the findings. How can the results inform healthcare policies or interventions to improve healthcare worker well-being and retention? This section should bridge the gap between research findings and real-world applications.
  • Quotation Integration: While providing illustrative quotes is valuable for understanding nurses' perspectives, ensure that the quotes are integrated smoothly into the text. You can introduce them more contextually, clarifying why each quote is relevant to the discussed subtheme or topic.
  • Conciseness: Some sections appear to contain redundant information. Review the content to eliminate repetition and ensure that each point contributes directly to understanding nurses' perspectives.
  • Policy Implications: Discussing potential policy implications or recommendations based on the findings would be beneficial towards the end of each subtheme. How can the issues raised by nurses be addressed at a policy level? This can provide a more practical dimension to the research.
  • Discussion of Limitations: It's essential to include a brief section discussing the study’s limitations. For example, mention any potential biases in the sample or limitations in the data collection methods.
  • Reference Citations: Ensure that you appropriately cite the sources of the quotations or statistics presented in the text, following a consistent citation style 
  • Formatting: Ensure that the formatting, such as font size and spacing, is consistent throughout the section.
  • Appendices: If there is an extensive amount of data, consider including an appendix with additional details, such as all the quotes from nurses, for readers who want to delve deeper into the responses.

Remember that this section aims to present the nurses' perspectives clearly and effectively, allowing readers to draw meaningful insights from the data. These suggestions should enhance the scientific quality and readability of the section.

Discussion:

  • Clarify Research Objectives: Begin the discussion section by restating the main research objectives and questions to remind readers of the study's purpose. This can help provide context for the subsequent discussion.
  • Highlight Novel Contributions: Clearly state the novel contributions of your study. What does your research bring to the existing body of knowledge on this topic? Emphasize how your study extends or enhances previous research, especially in COVID-19.
  • Link Findings to Research Questions: Ensure that the discussion of findings directly addresses the research questions you posed at the beginning of the study. Clearly state how each finding relates to answering these questions.
  • Consider Alternative Explanations: Discuss alternative explanations for your findings. Are there other factors that could explain the results you observed? Acknowledging potential alternative explanations demonstrates a thorough analysis.
  • Discuss Implications for Nursing Practice: Expand on the practical implications of your findings for nursing practice. How can the insights gained from your study be applied to improve working conditions for nurses? Provide specific recommendations based on your results.
  • Address Limitations in More Detail: While you mentioned limitations, consider expanding on them. Discuss how these limitations might have affected the study's outcomes and interpretations. Additionally, suggest ways to mitigate these limitations in future research.
  • Include a Section on Future Research: In addition to mentioning potential areas of future research in the introduction, dedicate a section specifically to discussing directions for future research based on your findings. What questions remain unanswered? How can future studies build upon your work?
  • Please provide a Summary. At the end of the discussion section, summarize the key takeaways from your study, including the main findings, their implications, and their potential contribution to the nursing profession, especially in the context of the COVID-19 pandemic.
  • Ensure Clarity and Flow: Ensure the discussion is well-structured and logically flows from one point to the next. Use clear headings and subheadings to guide readers through different aspects of the discussion.
  • Cite Relevant Literature: When discussing your findings, cite relevant literature to support your interpretations and comparisons with previous research. Make sure to include recent studies that provide context for your work, especially those related to the impact of COVID-19 on healthcare workers.
Comments on the Quality of English Language

  • Proofread and Edit: Carefully proofread the discussion section for grammatical errors, clarity, and coherence. Ensure that the writing is concise and free of jargon or unnecessary complexity.

Author Response

(Please see the attachment)

Dear Reviewer, We highly appreciate your insightful feedback. We've diligently revised our manuscript, addressing each point to enhance its quality and relevance. The improvements are detailed in the attached PDF. Your guidance has been crucial in improving our work, and we are eager to contribute meaningfully to nursing research. We appreciate the opportunity to refine our paper and look forward to any further suggestions. Thank you again for your valuable input.

Sincerely, The Corresponding Author

Reviewer 5 Report (New Reviewer)

Comments and Suggestions for Authors

Author Response

(Please see the attachment)

Dear Reviewer, We highly appreciate your insightful feedback. We've diligently revised our manuscript, addressing each point to enhance its quality and relevance. The improvements are detailed in the attached PDF. Your guidance has been crucial in improving our work, and we are eager to contribute meaningfully to nursing research. We appreciate the opportunity to refine our paper and look forward to any further suggestions. Thank you again for your valuable input.

Sincerely, The Corresponding Author

Round 2

Reviewer 2 Report (New Reviewer)

Comments and Suggestions for Authors

I believe that the significant revision of the thesis has improved readability a lot compared to the previous one.

Please summarize the research results on the table, focusing on the main contents. I don't think it's necessary to explain all the research results one by one. Because readers can see the data values or results in the table. Thank you.

Author Response

Please see the attachment. Thank you in advance.

Reviewer 4 Report (New Reviewer)

Comments and Suggestions for Authors

Greetings,

I am writing to express my gratitude for considering and incorporating my comments into your manuscript. Thank you once more, and I'm looking forward to reading your paper when it's finished.

Comments on the Quality of English Language

Minor corrections are needed. recommend a overall reading to check for minor errors spellings 

Author Response

Please see the attachment. Thank you in advance.

This manuscript is a resubmission of an earlier submission. The following is a list of the peer review reports and author responses from that submission.

Round 1

Reviewer 1 Report

Comments and Suggestions for Authors

Dear Authors,

thank you for the opportunity to review your manuscript.

The  topic is very interesting and important for the future of our Profession and discipline.

I read with attention your manuscript but in my opinion there are too many weakness in all of the text. I will highlight only some of them. The methodology chosen  is mixed methods but in reality the study is cross sectional. The mixed methods responds to other criteria and objectives.

The instruments used for data collection are questionnaires, no methods such as focus groups or interviews were chosen for collecting qualitative data. In the questionnaries used were two open questions, this is not (adapted to) the approach of a mixed methods study. There are too many inaccuracies in the section “methods” as well as errors in methodology.

The article is not well organised and confused in various sections. There are no ethical considerations.

.

Author Response

Please see the attachment. Thanks in advance!

Reviewer 2 Report

Comments and Suggestions for Authors

The research is very interesting. The methodology has been described in details and the results have been presented in details and legibly. Limitations have been taken into consideration in interpreting the results of this study. I recommend the paper for publication, however:

1. The aim of the study should be clearly indicated.

2. Some cited references relevant to the research should be translated into English (titles).

3. The paper needs minor revision (vocabulary errors and text editing).

Comments on the Quality of English Language

1. Minor editing of English language required. Ex. Table 12. Deviation between expectation and satisfaction more then 50% (than not then).

2. Some cited references relevant to the research should be translated into English (titles).

Author Response

(The authors gave the same response as above.)

Reviewer 3 Report

Comments and Suggestions for Authors

Dear authors,

Although I understand the pertinence of the manuscript, there are some points to be considered in terms of ethical considerations and methodology. Please refer to my comments and suggestions.

Introduction

Please reduce the length of descriptions of the Introduction and develop it more compactly.

Methods

Please describe the ethical considerations, such as obtaining IRB approval.

Instruments

Given the enormous number of participants, the readers would wonder about the extent to which the reliability of each instrument. Please describe the validity and reliability of all instruments that the present study utilized. 

Results

As mentioned above, given the enormous number of participants, it seems that their characteristics, namely sociodemographic characteristics in qualitative research, would not be the same as in quantitative research even though they gathered participants' information within the same research. It would be because not all participants who answered quantitative research questions would respond to qualitative inquiry. Please describe the participants' characteristics in the qualitative research.

Author Response

(The authors gave the same response as above.)
